# Transcriptional Activity of Metalloproteinase 9 (MMP-9) and Tissue Metalloproteinase 1 (TIMP-1) Genes as a Diagnostic and Prognostic Marker of Heart Failure Due to Ischemic Heart Disease

**DOI:** 10.3390/biomedicines11102776

**Published:** 2023-10-13

**Authors:** Dariusz Korzeń, Oskar Sierka, Józefa Dąbek

**Affiliations:** 1Provincial Specialist Hospital Megrez Sp. z o. o., Edukacji Street 102, 43-100 Tychy, Poland; 2Student Research Group at the Department of Cardiology, Department of Cardiology, Faculty of Health Sciences in Katowice, Medical University of Silesia in Katowice, Ziołowa Street 45/47, 40-635 Katowice, Poland; 3Department of Cardiology, Faculty of Health Sciences in Katowice, Medical University of Silesia in Katowice, Ziołowa Street 45/47, 40-635 Katowice, Poland

**Keywords:** genes, heart failure, coronary artery disease

## Abstract

The most common cause of heart failure (HF) is coronary artery disease (CAD). The aim of this study was to evaluate the transcriptional activity of the *metalloproteinase 9* (*MMP-9*) and *tissue metalloproteinase inhibitor 1* (*TIMP-1*) genes in a study group of patients with HF due to CAD and in the control group, as well as assess the transcriptional activity of the examined genes, taking into account the number of affected coronary arteries and the severity of heart failure. The study group consisted of a total of 150 (100%) patients. The material for the study was peripheral blood, and molecular tests were performed using the quantitative QRT-PCR technique. The transcriptional activity of the *MMP-9* gene was significantly higher in the group of patients with CAD and HF. It was also significantly higher with the progression of heart failure. *TIMP-1* gene transcriptional activity was significantly lower with the advancement of heart failure. The transcriptional activity of the *MMP-9* and *TIMP-1* genes differentiated the examined patients. The severity of HF, and a significant increase in the QRT-PCR transcriptional activity of the *MMP-9* gene with a simultaneous decrease in the activity of the *TIMP-1* gene, makes them useful diagnostic and prognostic markers in clinical practice.

## 1. Introduction

Heart failure (HF) is a clinical syndrome with symptoms and/or signs caused by a structural and/or functional abnormality of the heart, confirmed by elevated levels of natriuretic peptides and/or objective evidence of congestion in the small or large circulation [1]. The most common cause of the development of heart failure is coronary artery disease (CAD) developing based on atherosclerosis, in the course of which strictures and blood flow disorders form in the microcirculation. As a consequence, cardiomyocytes are damaged and not supplied with the nutrients and oxygen necessary for metabolic processes [2].

The heart is not homogeneous in terms of tissue construction, and the unique element of its structure are cells found nowhere else in the human body. Cardiomyocytes make up more than 90% of the mass of the heart muscle, but only about 30% of the number of all heart cells, while the remaining 70% are mainly fibroblasts and others, e.g., endothelial cells and macrophages [3,4]. The extracellular matrix is a connective tissue that forms the cytoskeleton between all cells that make up the wall of the heart. Its main ingredient is collagen synthesised by fibroblasts. The most common types of collagens in the heart—I and III—remain in a state of equilibrium and determine the mechanical effectiveness of its contraction and diastole [5]. In the extracellular matrix (ECM), there is a constant synthesis of new collagen fibres and their degradation. Most of these processes occur with the participation of metalloproteinases whose tissue inhibitors regulate their activity. Both types of proteins are secreted by, e.g., fibroblasts [6]. Collagen metabolism is also regulated by such factors as angiotensin II (ATII), transforming growth factor beta (TGF-β), aldosterone and tumour necrosis factor alpha (TNF-α) [7].

Matrix metalloproteinases (MMPs), also referred to as matrixins, are enzymes that hydrolyse extracellular matrix proteins. They belong to the class of endopeptidases dependent on zinc and calcium ions. The current classification of metalloproteinases distinguishes 25 endopeptidases, 23 of which are found in humans. MMPs, due to differences in substrate specificity and structure, have been divided into eight groups: collagenases, gelatinases, stromelysins, matrilysins, enamelisins, metalloelastases, membrane metalloproteinases and others, unclassified [8].

MMPs are inhibited by their tissue inhibitors (TIMPs), which are endogenous protein regulators. The TIMPs family (TIMP1–4) consists of proteins composed of 184 to 194 amino acids with a molecular weight of approximately 21 kDa. All TIMPs have two distinct domains, and the conformation of each domain is stabilized by disulfide bonds. Moreover, all TIMPs found in humans are approximately 40% identical in terms of amino acid sequences [9].

Adverse cardiac remodelling associated with excessive ECM remodelling contributes to the high morbidity and mortality of patients with heart disease. Elevated levels of MMPs in the ECM strongly correlate with left ventricular dysfunction in patients with heart failure [10]. After a myocardial infarction, many cell types begin to secrete MMPs, making them key regulators of the progression of cardiac remodelling. MMPs facilitate the degradation of damaged ECM and recruit more cells to remove dead cardiomyocytes and remodel the heart muscle [11,12]. Each phase contributes to temporary changes in MMPs levels. The initial, increased activity of pro-inflammatory cytokines released by damaged cardiomyocytes and inflammatory cells results in strong activation of MMPs; however, long-term stimulation contributes to an increase in the level of TIMPs, which ultimately leads to a decrease in the MMPs/TIMPs ratio, resulting in prolonged remodelling time of the damaged myocardium [13]. This process contributes to the development of heart failure.

With the advancement of technology, the identification of heart failure biomarkers has become more achievable, especially in the field of molecules related to cardiac ECM remodelling [14]. As already mentioned, the most common cause of HF is coronary artery disease and its acute complications, such as myocardial infarction. Molecular research can help in identifying patients at particular risk of developing heart failure in the course of CAD because, as a result of unfavourable changes caused by the disease, there are changes in the expression of various genes [15]. Both MMPs and their inhibitors (TIMPs) are interesting subjects of research on prognostic markers of the development of cardiovascular diseases, especially those accompanying the remodelling of the heart leading to its failure [16,17].

The aim of the study was to assess the transcriptional activity of the *metalloproteinase 9 (MMP-9)* and *tissue metalloproteinase 1 (TIMP-1)* genes in peripheral blood mononuclear cells (PBMCs) in a study group of patients with heart failure due to myocardial ischemia and in the control group, as well as to evaluate the transcriptional activity of the examined genes, taking into account the number of affected coronary arteries and severity of heart failure (LVEF%).

## 2. Materials and Methods

### 2.1. Materials

Before the commencement of the study, consent was obtained from the Bioethics Committee of the Medical University of Silesia in Katowice (KNW/0022/KB1/98/I/15/16, PCN/0022/KB1/36/21) for conducting the study, as well as written, informed consent of the participants to participate therein. The study involved 150 (100%) patients admitted successively to the Cardiology Department of the City Hospital and the Department of Cardiology, whose mean age was 65.72 ± 8.95 years. Among them were 80 (53.33%) with coronary artery disease and heart failure due to ischemic disease (C) in the stage of decompensation (NYHA III and IV), 40 (20.67%) with coronary artery disease excluded in coronary angiography (A) and 30 (20.00%) with non-insufficiency coronary artery disease (B). The inclusion criteria included age over 18 years, written informed consent to participate in the study, and the presence of documented coronary artery disease (significant and critical stenotic changes and occluded coronary arteries in coronary angiography, condition after coronary artery repair or coronary artery bypass surgery, and contractility disorders in transthoracic echocardiography) and clinically diagnosed decompensated heart failure in NYHA functional classes III-IV. The exclusion criteria included lack of consent of the patient to participate in the study, lack of confirmation of coronary artery disease by objective methods, heart failure caused by other causes, e.g., heart defects, shortness of breath caused by diseases of other organs and difficult contact with the patient.

The material for molecular tests was peripheral blood collected from the basilic vein of patients on the 1st day of hospitalization at the Department of Cardiology, simultaneously with other tests.

### 2.2. Methods

The study was carried out in two stages. Firstly, the transcriptional activity of genes associated with coronary heart disease and heart failure was assessed in the peripheral blood mononuclear cells using the qualitative technique of oligonucleotide microarrays, and the results obtained at this stage are the subject of another article. The second stage included the quantitative QRT-PCR assessment of the transcriptional activity of *metalloproteinase 9 (MMP-9)* and *tissue metalloproteinase 1 (TIMP-1)* genes selected by the Bland–Altman method as genes differentiating both healthy controls (coronary artery disease excluded in coronary angiography), as well as patients with coronary artery disease without heart failure from patients with coronary artery disease and heart failure in the above-mentioned cells.

#### 2.2.1. Isolation of Peripheral Blood Mononuclear Cells (PBMCs)

To obtain PBMC, blood was collected in tubes coated with ethylenediaminetetraacetic acid (EDTA) and then centrifuged in a Ficoll gradient to obtain peripheral blood mononuclear cells. Extraction of ribonucleic acid (RNA) from mononuclear cells was performed using TRIzol Reagent (Invitrogen), according to a modified method of Chomczyński and Sacchi [18].

#### 2.2.2. Nucleic Acid Extraction

RNA extracts were qualitatively assessed by electrophoresis in a 1% agarose gel with the addition of ethidium bromide (0.5 mg/mL) using the SUBMINI apparatus. After separation, the electropherograms were evaluated using a UV transluminator (λ = 260 nm).

Total RNA extracts were quantified by spectrophotometric measurement of RNA concentration (Gene Quant II, Pharmacia). The spectrophotometric evaluation of the extracts included the measurement of absorbance at the wavelengths of 230, 260, 280 and 320 nm, and then the determination of the A260/A280 ratio and determination of the protein content. The absorbance value at the wavelength of 260 nm was used to calculate the RNA concentration, assuming that the measurement result in a cuvette with an optical path of 1 cm equal to 1 OD260 corresponds to a concentration of 40 mg of RNA in 1 cm^3^ of the extract.

#### 2.2.3. Real-Time Quantitative Polymerase Chain Reaction (QRT-PCR)

Transcriptional activity of the studied genes was assessed using commercial TaqMan Gene Expression Assays kits labelled at the 5’ ends with the fluorescent dye FAM (6-carboxy-fluorescein), and at the 3’ end with a non-fluorescent quencher. The number of mRNA molecules of the tested genes MMP-9, TIMP-1, β-actin and GAPDH was determined based on the kinetics of the QRT-PCR reaction using the StepOnePlus sequence detector (Applied Biosystems) and a kit containing the fluorescent dye ROX TaqMan™ RNA-to-CT ™ (Thermo Fisher Scientific, Waltham, MA, USA) 1-Step Kit. QRT-PCR was performed in a single-step reaction mixture containing TaqMan RT-PCR Mix (2×) (AmpliTaq Gold DNA Polymerase), TaqMan RT Enzyme Mix (40×) (ArrayScript™ UP) (Thermo Fisher Scientific, Waltham, MA, USA), TaqMan Gene Expression Assay primers and probe mix (Applied Biosystems), RNA template and non-pyrogenic water.

Simultaneously with the studied genes, commercially available DNA standards of the β-actin gene were amplified. The reaction mixes for the amplification of β-actin gene DNA standards contained 2× TaqMan^®^ Gene Expression Master Mix (AmpliTaq Gold^®^ DNA Polymerase, dNTP mix, reference ROX), beta-actin cDNA template and non-pyrogenic water (Thermo Fisher Scientific, Waltham, MA, USA).

The reverse transcription reaction was carried out at 500 °C for 30 min. (reverse transcription), 950 °C for 15 min. (polymerase activation), 45 cycles of a two-step reaction—94 °C for 15 s (denaturation) and 60 °C for 1 min (hybridization and extension) and a final extension of the amplification products at 72 °C for 10 min.

Based on the obtained results, a standard curve was obtained for each analysis. Based on the plotted standard curve, the StepOnePlus sequence detector determined the number of copies of the analysed genes.

The expression of the studied genes was inferred based on the number of mRNA copies per 1 µg of total RNA.

### 2.3. Statistical Analysis

The obtained results were collected in an Excel spreadsheet and exported to Statistica 12 (StatSoft Poland, Kraków, Poland). The mean values, standard deviation (SD), median (Me) and quartile range (IQR) were calculated. The normality of the distribution of the examined parameters was checked with the Shapiro–Wilk test. Since the examined parameters did not show the features of normal distribution, non-parametric tests were used in the statistical analysis. To compare the analysed parameters in the study groups, the Kruskal–Wallis ANOVA and Mann–Whitney U tests were used. The significance level of *p* < 0.05 was considered statistically significant.

## 3. Results

The characteristics of the study group, taking into account sex and age, are presented in Table 1.

The majority of the study group were men (133; 88.67%), and the average age of the respondents was 65.72 ± 8.95 years.

The characteristics of the examined patients, taking into account the left ventricular ejection fraction, are presented in Table 2.

Taking into account the division following the *ESC guidelines of 2021 on the diagnosis and treatment of acute and chronic heart failure*, the largest percentage (70; 46.67%) of the study group was patients with an ejection fraction ≥ 50% (A, B) being the control for patients with coronary artery disease and heart failure (C).

Figure 1 presents the results of a comparison of the transcriptional activity of the *metalloproteinase 9 (MMP-9)* gene in patients with coronary-angiography-excluded coronary artery disease (A), patients with coronary artery disease without heart failure (B) and patients with coronary artery disease and heart failure (C) evaluated by QRT-PCR in peripheral blood mononuclear cells.

Transcriptional activity of the *metalloproteinase 9 (MMP-9)* gene was significantly higher in the group of patients with coronary artery disease and heart failure compared to patients with coronary artery disease without heart failure and those with coronary artery disease excluded in coronary angiography.

Figure 2 presents the results of the comparison of the transcriptional activity of the *tissue metalloproteinase inhibitor 1 (TIMP-1)* gene in patients with coronary angiography excluding coronary artery disease (A), patients with coronary artery disease without heart failure (B) and patients with coronary artery disease and heart failure (C) evaluated by QRT-PCR in peripheral blood mononuclear cells.

Transcriptional activity of the *tissue metalloproteinase inhibitor 1 (TIMP-1)* gene was significantly higher in the group of patients with coronary artery disease without heart failure and in patients with coronary artery disease and heart failure compared to coronary artery disease excluded in coronary angiography.

Comparison of transcriptional activity of the *metalloproteinase 9 (MMP-9)* and *tissue inhibitor of metalloproteinases 1 (TIMP-1)* genes in patients with coronary artery disease without heart failure (B) and in patients with coronary artery disease and heart failure (C) by the number of affected coronary arteries are shown in Figure 3 and Figure 4.

The analysis showed no differences in the transcriptional activity of the examined genes depending on the number of affected coronary arteries.

Results of comparisons of the transcriptional activity of the *metalloproteinase 9 (MMP-9)* and *tissue inhibitor of metalloproteinases 1 (TIMP-1)* genes using the Kruskal–Wallis test in patients with coronary angiography excluded coronary artery disease (A), patients with coronary artery disease without heart failure (B) and patients with coronary artery disease and heart failure (C) are shown in Figure 5 and Figure 6.

With the advancement of heart failure (decrease in left ventricular ejection fraction), an increase in the transcriptional activity of the *metalloproteinase 9 (MMP-9)* gene was found.

With the advancement of heart failure (decrease in left ventricular ejection fraction), a decrease in the transcriptional activity of *tissue metalloproteinase inhibitor 1 (TIMP-1)* gene was found.

## 4. Discussion

The analysis of the transcriptional activity of genes using QRT-PCR made it possible to separate genes differentiating patients with heart failure in the course of coronary artery disease developing based on atherosclerosis from the control group. The differentiating genes within the indicated criteria were genes for *metalloproteinase type 9 (MMP-9)* and *tissue inhibitor of metalloproteinases 1 (TIMP-1)*, which may indicate their important role in the development of atherosclerosis and its progression and heart failure.

The subcellular structure of the heart includes many proteins (e.g., collagens, laminins, fibronectin) and other substances. Collagen is the most abundant protein in the extracellular matrix (ECM) and is the scaffold that provides cardiac muscle fibres with three-dimensional structure and tensile strength. In both heart failure and other cardiovascular diseases, the heart muscle is remodelled to maintain its function. The redevelopment includes, e.g., breakdown of the collagen network with the participation of MMPs, including MMP-9 [5]. This metalloproteinase, originally referred to as 92-kDa collagenase type IV or gelatinase B, plays a major role in the degradation of the ECM in a wide range of physiological and pathophysiological processes, including the remodelling of myocardial tissue since embryonic development. MMP-9 is present in the developing human heart tissue and is expressed between the 16th and 18th day of embryogenesis [19]. It is also suggested that MMP-9 plays a significant role in neovascularization through proteolytic degradation of proteins in the basal lamina of blood vessels and the release of a biologically active form of vascular endothelial growth factor [20]. In pathophysiological conditions, the level of MMP-9 increases during wound healing, as well as in inflammatory processes, including arthritis, diabetes and cancer [21]. The level of MMP-9 also increases rapidly in cardiovascular diseases such as arterial hypertension, atherosclerosis and myocardial infarction, and a significant number of publications on MMP-9 emphasize its importance as an important and useful marker in diagnosis and estimation of prognosis [22]. A study by Braiek B. et al. showed that blood MMP-9 levels are elevated in patients with coronary artery disease compared to healthy individuals [23]. An increased level of MMP-9 in patients with coronary artery disease as compared to patients without this disease was also demonstrated by Xu Y. et al. [24]. Studies conducted by Jong G. et al. to assess serum concentration and activity of MMP-9 in a group of twenty-eight patients with myocardial infarction (MI) without myocardial infarction and twenty-seven patients with MI and heart failure showed that both tested parameters related to MMP-9 increased significantly (*p* < 0.01) in the group of patients with MI and failure compared to patients with MI without failure [25]. Medeiros N. et al. showed that the level of MMP-9 seems to be a biomarker of late fibrosis and severe heart remodelling in patients with cardiovascular diseases, i.e., heart failure [26].

The results of a study by Georg J. et al. suggested that it would be beneficial to lower MMP-9 levels to reduce side effects. Standard pharmacotherapy can help with this, as several drugs used to treat heart failure and other conditions show activity against MMP-9. These include drugs targeting the renin–angiotensin–aldosterone system, including ACE inhibitors, angiotensin receptor blockers and aldosterone receptor antagonists, which have been confirmed in both animal and human models [27].

Studies have also shown that ACE inhibitors reduced the activity of MMP-9 by directly binding to its active site, and statins stabilize atherosclerotic plaque by inhibiting several MMPs, including MMP-9 [28,29]. To date, several selective MMP-9 inhibitors have been developed and successfully tested in animal models. In case of heart damage, salvian acid, a selective inhibitor of MMP-9, prevented heart remodelling in spontaneously hypertensive rats [30]. Although selective inhibition of metalloproteinases is still in the realm of concept, Scannevin R. et al. have identified a highly selective compound that inhibits the activation of the MMP-9 zymogen and the subsequent production of the catalytically active enzyme. The discovered compound JNJ0966 did not affect the catalytic activity of MMP-1, MMP-2, MMP-3, MMP-9 or MMP-14 and did not inhibit the activation of the highly related zymogen MMP-2. The molecular basis of the reported activity was characterized as the interaction of JNJ0966 with a structural pocket near the MMP-9 zymogen cleavage site, distinct from the catalytic domain. The compound JNJ0966 was effective in reducing disease severity in a mouse model of experimental autoimmune encephalomyelitis, suggesting its future therapeutic potential [31]. Other studies in this area are also conducted [32].

The balancing effect for MMP-9 is shown by TIMP-1, which can also inhibit other MMPs except those associated with the cell membrane. Increased levels of TIMP-1 in tissues and plasma correlate with the degree of myocardial fibrosis and its diastolic dysfunction [33]. The above-mentioned researchers, Braiek B. et al., also showed a statistically significant decrease in TIMP-1 in patients with coronary artery disease compared to patients without said disease [23].

According to Creemers E. et al., low levels of TIMP-1 contributed to left ventricular remodelling after experimentally induced myocardial infarction [34]. Sundström J. et al. noted that among healthy men, higher plasma TIMP-1 levels were associated with increased left ventricular diastolic dimensions and increased wall thickness [35]. Frantz S. et al. showed that changes in the level of TIMP-1 were associated with the severity of heart failure defined by the NYHA class, the presence of jugular venous dilatation, peripheral oedema, the use of diuretics, and the size and end-diastolic volume of the left ventricle [33]. In addition, in the cited study, the baseline TIMP-1 level turned out to be a strong predictor of death from any cause at 24-month follow up. Patients with the highest tertile of TIMP-1 values had an eightfold increased risk of death. TIMP-1 levels also showed additional prognostic information when added to the clinical summary set and biochemical descriptors of heart failure, including NT-proBNP. In particular, the combination of high TIMP-1 and high NT-proBNP levels was associated with a particularly poor prognosis. One likely explanation for this additional deleterious effect could be that NT-proBNP mainly reflects worsened stress conditions, while TIMP-1 reflects changes in the extracellular matrix in HF patients [36]. A study by Fouda M. et al., which evaluated the diagnostic efficacy of fibronectin, TIMP-1 and CK-MB in heart failure, showed that fibronectin was the most significant biomarker in differentiating HF patients from healthy controls (AUC = 0.850) (*p* < 0.001), followed by TIMP-1 (AUC = 0.74) and CK MB (AUC = 0.660). Unfortunately, the main limitation of the study was the small number of participants [37].

TIMP-1 has also been evaluated in studies as a biomarker of myocardial fibrosis [38,39]. Unfortunately, so far it has not been possible to fully use its potential in this role. Despite the consistent relationship between TIMP-1 levels and fibrosis, it is not known whether TIMP-1 directly promotes tissue fibrosis or whether its increase is an accidental change in this process. A study by Takawale A. et al. also showed that TIMP-1 is involved in inducing myocardial fibrosis, independently of its function of inhibiting MMPs by mediating the interaction between fibroblast membrane proteins and the CD63 receptor and β1 integrin. With this new information on the role of TIMP-1 in the process of fibrosis, the discovery of targeted therapy may prove beneficial in limiting this process in patients with heart disease [40]. The effect of pharmacotherapy on TIMPs levels is not well studied. In the presented meta-analysis, Ferretti G. et al. showed that patients treated with statins exhibited a significant decrease in plasma TIMP-1 levels (SMD: −0.30, 95% CI: −0.56 to −0.03, *p* = 0.029) [41].

The statistical analysis of the results of our research showed no significant differences in the transcriptional activity of the examined genes depending on the number of diseased coronary arteries. The above was confirmed by the studies of Braiek B. et al., who showed no differences in the level of TIMP-1 depending on the number of diseased coronary arteries [23]. Different results were obtained by Mahmoodi K. et al., who observed an increase in the concentration of circulating MMP-9 in patients with three-vessel coronary artery disease compared to patients in the control group without coronary atherosclerosis [42]. Moreover, a study by Nishiguchi T. et al. showed a higher level of MMP-9 in patients with STEMI after stenting than in the control group [43].

The performed QRT-PCR analysis in the study group of patients with heart failure showed a statistically significant increase in the transcriptional activity of the *metalloproteinase 9 (MMP-9)* gene and a decrease in the activity of the *tissue metalloproteinase inhibitor 1 (TIMP-1)* gene with its advancement (decrease in the ejection fraction). Similar results were obtained in the work of Yan A. et al., who showed that the increased concentration of MMP-9 in plasma correlated with lower left ventricular ejection fraction [44]. An increase in the concentration of metalloproteinases (MMPs) and a decrease in their tissue inhibitors (TIMPs) depending on the left ventricular ejection fraction was also demonstrated by Kobusiak-Prokopowicz M. et al. [45]. On the other hand, a study by Braiek B. et al. showed that the levels of MMP-9 in the blood are elevated in patients with coronary artery disease compared to healthy people [23].

Limitations of the study include the small number of participants, as well as the inequality between the number of women and men. It is worth emphasizing here the possible causes of the observed imbalances. Firstly, in accordance with the assumptions and adopted methodology, patients were included in the study in the order in which they presented to the cardiology department and/or clinic due to decompensation of heart failure in NYHA class III-IV. Secondly, as research by Alefan Q. et al. showed, women have a higher level of compliance than men with regard to the use of therapy, and this is most likely why not as many of them were admitted to the cardiology department and/or clinic due to exacerbations of heart failure—which was the inclusion criterion for the study [46]. Thirdly, the small number of females included in the study could also be related to problem with availability of diagnostic and therapeutic procedures for women. This disturbing situation was confirmed by the research of Cader F. et al., who showed significant differences in the availability of cardiological diagnostic and therapeutic procedures for women and men in Europe [47]. As mentioned earlier, only patients with documented coronary artery disease (significant and critical changes in stenosis and occlusion of coronary arteries in coronary angiography, condition after arterial repair, coronary artery bypass surgery and contractility disorders in transthoracic echocardiography) and clinically diagnosed decompensated heart failure in NYHA functional classes III-IV were included in the study group. An incomplete diagnosis made at earlier stages of treatment excluded a given patient from the study group.

Nevertheless, the study confirms the need and allows for further exploration of the usefulness of testing the transcriptional activity of genes in peripheral blood mononuclear cells, especially in terms of categorizing patients into different risk groups, as well as in terms of their usefulness as diagnostic and prognostic markers of the course of heart failure caused by coronary artery disease. Studies of the transcriptional activity of selected genes in the mentioned cells may also be used in the future to develop new therapeutic methods and personalize the therapies used, as well as become an indicator of patients’ compliance with given by their health providers.

Thanks to multicentre studies of the molecular basis of heart failure, a full picture of the causes of this disease is slowly beginning to emerge. Furthermore, with the progress of research methods and more thorough knowledge of various pathways leading to heart failure, the possibility of searching for new methods of diagnosis, treatment and prevention of this disease increases. Further research should include a more accurate assessment of genes and their expression products, especially in terms of potential starting points for the development of the disease.

## 5. Conclusions

Transcriptional activity of *metalloproteinase 9 (MMP-9)* and *tissue inhibitor of metalloproteinase 1 (TIMP-1)* genes differentiates healthy subjects (with coronary artery disease excluded in coronary angiography) from patients with coronary artery disease without heart insufficiency and patients with coronary artery disease complicated by heart failure. The severity of coronary artery disease does not correlate with the transcriptional activity of the examined genes, in contrast to the severity of heart failure (decrease in left ventricular ejection fraction). The significant increase in QRT-PCR transcriptional activity of the *metalloproteinase 9 (MMP-9)* gene with a simultaneous decrease in the activity of *tissue inhibitor of metalloproteinases 1 (TIMP-1)* gene in the study group of patients with coronary artery disease and heart failure makes them useful diagnostic and prognostic markers in clinical practice.

## Figures and Tables

**Figure 1 biomedicines-11-02776-f001:**
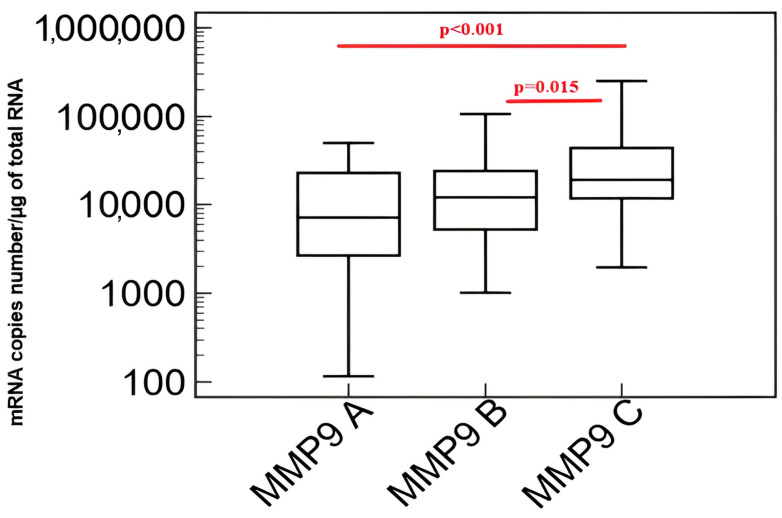
Transcriptional activity of the *metalloproteinase 9 (MMP-9)* gene in the study group. Explanation of abbreviations: A—patients with coronary artery disease excluded in coronary angiography, B—patients with coronary artery disease without heart failure, C—patients with coronary artery disease and heart failure. Red color indicates the occurrence of statisticaly important differences between studied groups.

**Figure 2 biomedicines-11-02776-f002:**
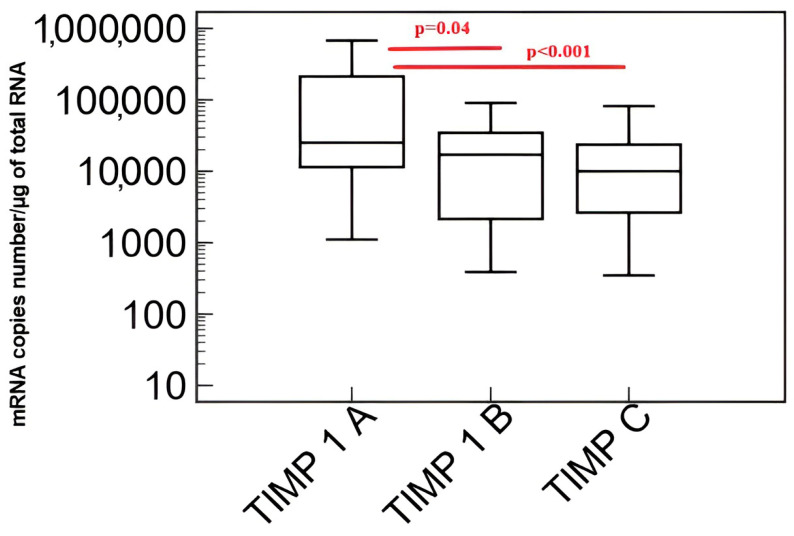
Transcriptional activity of the *tissue metalloproteinase inhibitor 1 (TIMP-1)* gene in the study group. Explanation of abbreviations: A—patients with coronary artery disease excluded in coronary angiography, B—patients with coronary artery disease without heart failure, C—patients with coronary artery disease and heart failure. Red color indicates the occurrence of statisticaly important differences between studied groups.

**Figure 3 biomedicines-11-02776-f003:**
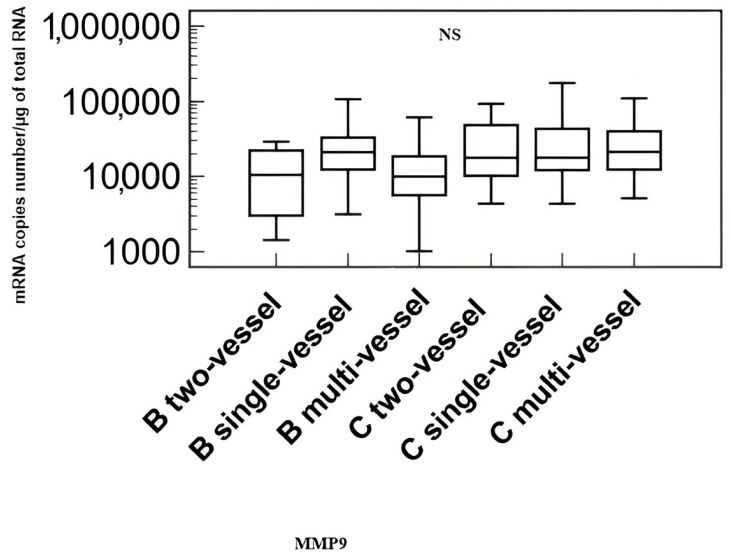
Transcriptional activity of the *metalloproteinase 9 (MMP-9)* gene in the study group, taking into account the number of affected coronary arteries. Explanation of abbreviations: B—patients with coronary artery disease without heart failure, C—patients with coronary artery disease and heart failure, NS – non significant.

**Figure 4 biomedicines-11-02776-f004:**
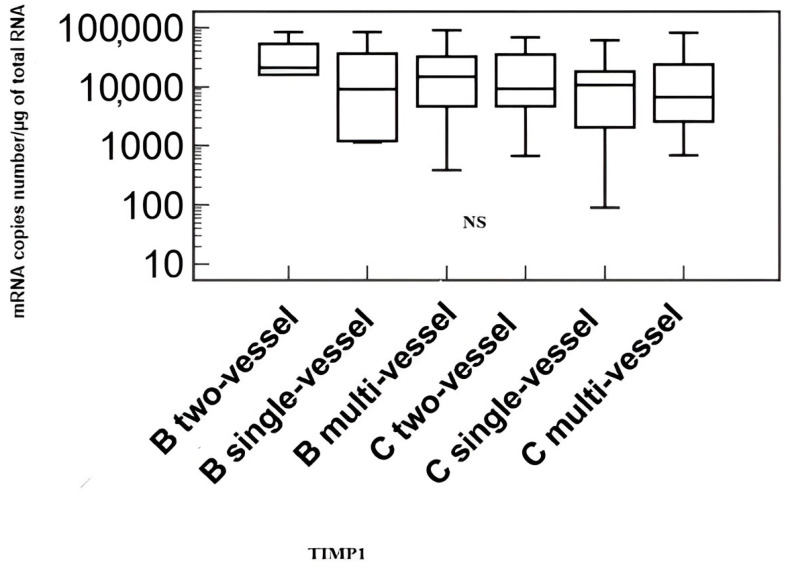
Transcriptional activity of the *tissue metalloproteinase inhibitor 1 (TIMP-1)* gene in the study group of patients, taking into account the number of diseased coronary arteries. Explanation of abbreviations: B—patients with coronary artery disease without heart failure, C—patients with coronary artery disease and heart failure, NS – non significant.

**Figure 5 biomedicines-11-02776-f005:**
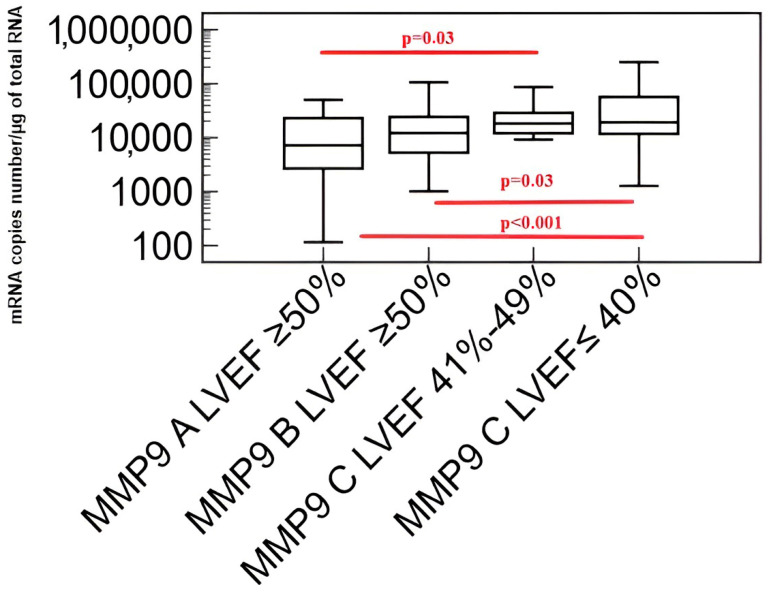
Transcriptional activity of the *metalloproteinase 9 (MMP-9)* gene in the study group. Explanation of abbreviations: A—patients with coronary artery disease excluded in coronary angiography, B—patients with coronary artery disease without heart failure, C—patients with coronary artery disease and heart failure, LVEF—left ventricular ejection fraction. Red color indicates the occurrence of statisticaly important differences between studied groups

**Figure 6 biomedicines-11-02776-f006:**
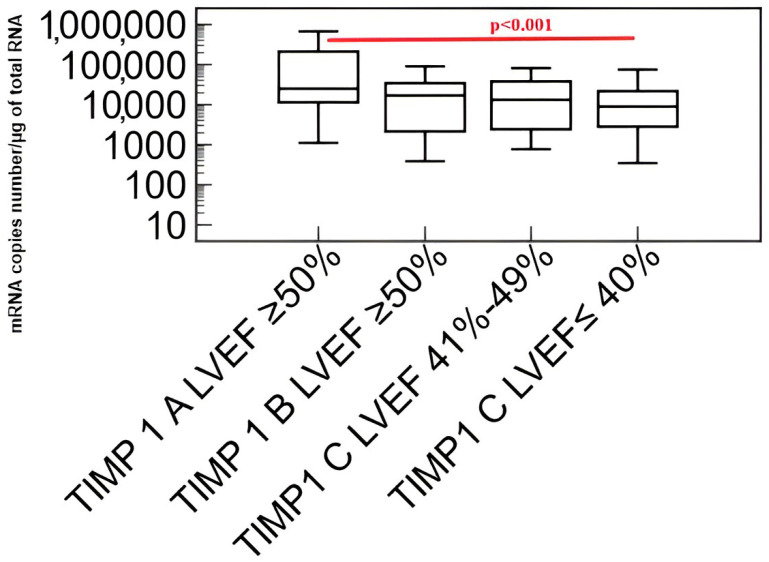
Transcriptional activity of the *tissue metalloproteinase inhibitor 1 (TIMP-1)* gene in the study group of patients. Explanation of abbreviations: B—patients with coronary artery disease without heart failure, C—patients with coronary artery disease and heart failure, LVEF—left ventricular ejection fraction. Red color indicates the occurrence of statisticaly important differences between studied groups

**Table 1 biomedicines-11-02776-t001:** Characteristics of the study group taking into account gender and age.

**Variable**		**Studied Group (150; 100%)**								
		**ABC** ***n* = 150 (100%) **		**A** ***n* = 30 (20.00%)**		**B** ***n* = 40 (26.67%) **		**C** ***n* = 80 (53.33%) **		** *p* **
**Sex**	**Woman**	*n* = 17 (11.33%)		*n* = 15 (50.00%)		*n* = 1 (2.5%)		*n* = 1 (1.25%)		
	**Man**	*n* = 133; (88.67%)		*n* = 15 (50.00%)		*n* = 39 (97.5%)		*n* = 79 (98.75%)		
**Variable**		**X** **±SD**	**Me** **IQR**	**X** **±SD**	**Me** **IQR**	**X** **±SD**	**Me** **IQR**	**X** **±SD**	**Me** **IQR**	
**Age [years]**		65.72 ± 8.95	65.00 13.00	62.77 ± 9.29	64.50 11.00	64.67 ± 10.45	63.00 18.00	67.35 ± 7.67	66.00 11.50	NS

Explanation of abbreviations: A—patients with coronary artery disease excluded in coronary angiography, B—patients with coronary artery disease without heart failure, C—patients with coronary artery disease and heart failure, n—sample size, X—mean, ±SD—standard deviation, Me—median, IQR—quartile range, *p*—statistical significance, NS—not significant.

**Table 2 biomedicines-11-02776-t002:** Characteristics of the examined patients, taking into account the left ventricular ejection fraction (*classification by the 2021 ESC guidelines for the diagnosis and treatment of acute and chronic heart failure*).

Studied Group (150; 100%)	Left Ventricular Ejection Fraction (LVEF) in %					
	HFrEF (≤40%)		HFmrEF (41–49%)		HFpEF (≥50%)	
Sample Size (*n* and %) of a Given Group	*n*	%	*n*	%	*n*	%
**ABC** ***n* = 150; 100% **	60	40.00%	20	13.33	70	46.67%
**A** ***n* = 30; 100% **	0	0.00%	0	0.00%	30	100%
**B** ***n* = 40; 100% **	0	0.00%	0	0.00%	40	100%
**C** ***n* = 80; 100% **	60	75.00%	20	25.00%	0	0.00%

Explanation of abbreviations: A—patients with coronary artery disease excluded in coronary angiography, B—patients with coronary artery disease without heart failure, C—patients with coronary artery disease and heart failure, n—sample size, LVEF—left ventricular ejection fraction, HFrEF—heart failure with reduced ejection fraction, HFmrEF—heart failure with mildly reduced ejection fraction, HFpEF—heart failure with preserved ejection fraction.

## Data Availability

All data can be obtained from the corresponding author upon reasonable request.

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
