# Peer review of "Transcriptional Activity of Metalloproteinase 9 (MMP-9) and Tissue Metalloproteinase 1 (TIMP-1) Genes as a Diagnostic and Prognostic Marker of Heart Failure Due to Ischemic Heart Disease"

_biomedicines, 2023, doi:10.3390/biomedicines11102776_

Round 1
Reviewer 1 Report
In the current manuscript, the authors evaluated the mRNA expression of MMP-9 and TIMP-1 in the peripheral blood mononuclear cell of CAD with or without heart failure and determined the possibility of MMP9 and TIMP1 as diagnostic and prognostic markers of ischemic heart failure. The major concern for this manuscript is the novelty of the concept of these two genes as diagnostic markers in ischemic heart failure as these two genes are well-known to be associated with heart failure in humans and mice (PMID: 16830792, 15309700, 32643898). Here are several aspects that could help to improve the manuscript.
-One of the major concerns is about the gender of the group B and C. The n number for females is only equal to 1, so the result from the current manuscript should mostly indicate the features of males.
-Although the authors already measured the mRNA expression of MMP9 and TIMP1 in subgroups with different LVEF, what’s the association between the mRNA expression of the genes with LVEF?
-The whole manuscript uses “transcriptional activity” to indicate the mRNA level of two genes. However, the assay used in this manuscript was measuring the mRNA expression in the blood mononuclear cell. It’s not so accurate to use the “transcriptional activity”.
-As MMP9 and TIMP1 are secreted proteins, what’s the protein expression of these two proteins in the peripheral blood serum of the patients?
Author Response
Dear Reviewer,
As requested we have considered all your concerns and improved our manuscript. We hope that the corrections introduced into the manuscript body are sufficient and allows our manuscript to be published. All changes are described in the attached file,
Yours faithfully,
Dariusz KorzeÅ„, Oskar Sierka & Józefa DÄ…bek

Reviewer 2 Report
The article "Transcriptional activity of metalloproteinase 9 (MMP-9) and tissue metalloproteinase 1 (TIMP-1) genes as a diagnostic and prognostic marker of heart failure due to ischemic heart disease" provides a comprehensive overview of the transcriptional activity of MMP-9 and TIMP-1 genes in patients with heart failure due to ischemic heart disease. The study design, methods, and results are presented clearly. I have very few comments (in no order of magnitude)
· Line 39, Add suitable reference for the percentage of cardiomyocytes and non-cardiomyocytes in humans
· In the introduction, please include references for the cellular source for the secretion of MMPs and TIMPs in the heart during remodeling
· Elaborate further the diagnostic potential of this study in clinical practice in the discussion
· If the fibrosis data available for the patient groups selected, please include and co-relate this with the transcriptional activity of MMP9 and TIMP1
Author Response

(The authors gave the same response as above.)

Round 2
Reviewer 1 Report
The major concern for this manuscript is the novelty of the concept and I don't think that there is a significant improvement in the revised version.